# EEG-representational geometries and psychometric distortions in approximate numerical judgment

**Stefan Appelhoff**[1,2,3]*, **Ralph Hertwig**[2], **Bernhard Spitzer**[1,2,3]

**1** Research Group Adaptive Memory and Decision Making, Max Planck Institute for Human Development, Berlin, Germany, **2** Center for Adaptive Rationality, Max Planck Institute for Human Development, Berlin, Germany, **3** Max Planck Dahlem Campus of Cognition, Max Planck Institute for Human Development, Berlin, Germany

* appelhoff@mpib-berlin.mpg.de

**Data Availability Statement:** All data is available on GIN (see https://doi.org/10.12751/g-node. lir3qw). The analysis code and experiment code are hosted on GitHub and archived on Zenodo (see

## Abstract

When judging the average value of sample stimuli (e.g., numbers) people tend to either over- or underweight extreme sample values, depending on task context. In a context of overweighting, recent work has shown that extreme sample values were overly represented also in neural signals, in terms of an anti-compressed geometry of number samples in multivariate electroencephalography (EEG) patterns. Here, we asked whether neural representational geometries may also reflect a relative underweighting of extreme values (i.e., compression) which has been observed behaviorally in a great variety of tasks. We used a simple experimental manipulation (instructions to average a single-stream or to compare dual-streams of samples) to induce compression or anti-compression in behavior when participants judged rapid number sequences. Model-based representational similarity analysis (RSA) replicated the previous finding of neural anti-compression in the dual-stream task, but failed to provide evidence for neural compression in the single-stream task, despite the evidence for compression in behavior. Instead, the results indicated enhanced neural processing of extreme values in either task, regardless of whether extremes were over- or underweighted in subsequent behavioral choice. We further observed more general differences in the neural representation of the sample information between the two tasks. Together, our results indicate a mismatch between sample-level EEG geometries and behavior, which raises new questions about the origin of common psychometric distortions, such as diminishing sensitivity for larger values.

## Introduction

When making decisions about magnitudes such as numbers, people tend to distort sample information away from its true value. A commonly observed distortion is a compression of magnitude, where extreme (or outlying) samples receive relatively less weight than prescribed by a linear and, according to common interpretation, normative transformation of objective

https://doi.org/10.5281/zenodo.6411287 and
https://doi.org/10.5281/zenodo.6411313).

**Funding:** This work was supported by a European
Research Council Consolidator Grant ERC-2020-
COG-101000972 (BS) and by DFG Grant SP 1510/
6-1 (BS). The funders had no role in study design,
data collection and analysis, decision to publish,or
preparation of the manuscript.

**Competing interests:** The authors have declared
that no competing interests exist.

into psychological or subjective values [1–6]. However, in some task contexts, the opposite type of distortion has been observed, that is, an anti-compression, where extreme or outlying samples are overweighted [7–12].

Using electroencephalographic (EEG) recordings and multivariate representational similarity analysis (RSA), recent work has identified a potential neural signature of such psychometric distortions. During processing of symbolic number samples, multivariate EEG patterns have been characterized by a "numerical distance effect", where the representational similarity of, for instance, "4" and "5" is higher than that of "4" and"6", which in turn is larger than that of "3" and "7", and so forth [9,10,13–15]. Intriguingly, in a multi-sample decision task that promoted anti-compression of number samples in behavior, the "neural numberline" underlying the numerical distance effect was found to be anti-compressed as well [9,10,13]. The findings suggested that behaviorally relevant distortions in multi-sample decisions may occur already when the individual samples are being encoded one by one.

However, such "neurometric" signature of psychometric distortion has thus far only been reported in tasks that promoted anti-compression, that is, a selective overweighting of extreme (outlying) sample values in behavior. A much more common behavioral finding in other task contexts is a *compression* of magnitude, where extreme values are underweighted, for instance, in psychophysical tasks [2,16,17], in studies of numerical cognition [18–21], and in behavioral economics experiments [5,22,23]. To what extent large-scale neural patterns, as recorded with human EEG, may also reflect psychometric compression is still unknown.

In the present study, we capitalized on recent progress in understanding the experimental factors that may mediate whether people compress or anti-compress magnitudes in decision making [see also 24]. Specifically, in a recent behavioral study, we found compression when participants judged the average of a single stream, but anti-compression when they compared dual streams of number samples [7]. Here, we adopted this experimental manipulation to examine the neural signatures of compressive (as compared to anti-compressive) number processing in multivariate EEG patterns.

Behaviorally, the results confirmed a compression of numerical values in the single-stream task, and an anti-compression of the same values in the dual-stream task. In neural signals, we replicated the finding of an anti-compressed number representation in the dual-stream task. Surprisingly, however, we found no evidence for neural compression in the single-stream task. Instead, we observed more general differences in the neural representation of the sample information. Whereas in the dual-stream task, the samples' neural geometry predominantly reflected their abstract magnitude, the single-stream task was associated with a more direct, non-quantitative representation of the concrete sample stimuli. The results relativize the diagnosticity of sample-level EEG-metrics for psychometric distortions. They also suggest a default pattern in EEG signals, namely, enhanced processing of extreme sample values, regardless of whether these values are over- or underweighted in subsequent behavior.

## Results

Participants (n = 30) performed two different variants of a sequential integration task where they observed sequences of ten digits (ranging between 1 and 9, colored in red or blue; Fig 1A). In the single-stream variant ("averaging" task), participants were asked to report whether the average of all ten number samples (regardless of color) was larger or smaller than 5. In the dual-stream variant ("comparison" task), they were asked to indicate whether the red samples had a higher average value than the blue samples, or vice versa. Intuitively, the latter task is cognitively more demanding [see also 7] as it requires evaluating the sample values according to their respective colors, whereas in the single-stream task, the colors can be ignored.

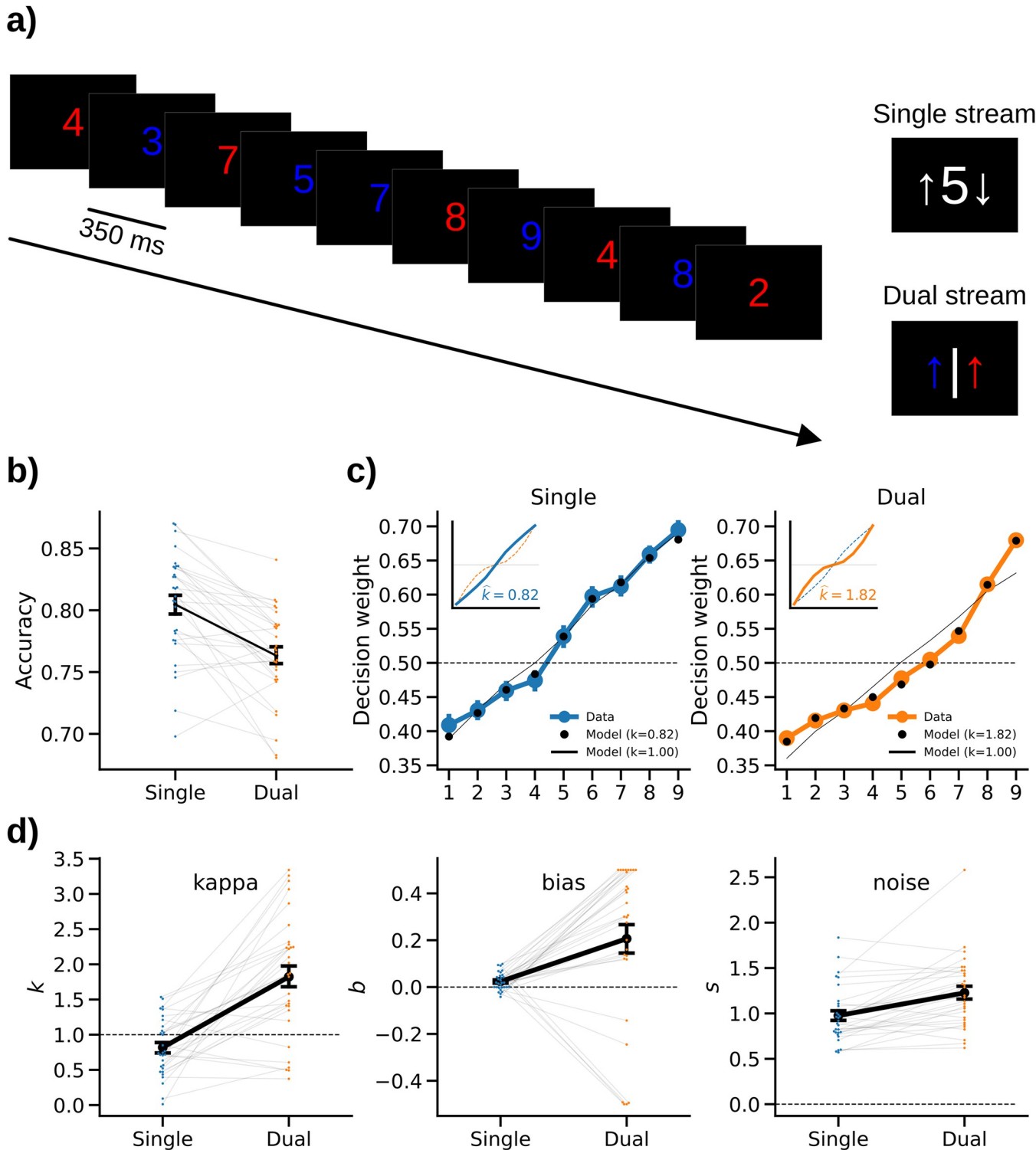

**Fig 1. Experimental paradigm and behavioral results. a)** Each participant performed two variants of a sequential number integration task. *Left*: In both variants, participants viewed a stream of ten rapidly presented digits (five in red and five in blue color; in random serial order) drawn from 1 to 9 (uniform random). In the single-stream averaging task, participants were asked to judge whether the average value of all ten samples was higher or lower than 5 (ignoring the samples' colors). In the dual-stream comparison task, participants were asked to compare the average value of the red samples against that of the blue samples and to report which was larger. *Right*: In both tasks, the response mapping onto left/right button presses was randomized across trials and cued only after sample presentation, in order to avoid motor preparation confounds. **b)** Mean accuracy (proportion correct choices) in the two tasks. **c)** Decision weights of number values in the single-stream (left) and dual-stream (right) tasks. Inset plots illustrate the shape of distortion implied by the best-fitting *k* (see panel *d*)

according to model Eq 1, with *b* set to 0 for visual comparability between task conditions. Thin dotted lines show the shape of distortion in the other task for comparison. **d)** Parameter estimates from fitting our psychometric model to the empirical choice data (cf. *c*). Error bars in all panels show SE.

## Behavioral results

As expected, mean choice accuracy (Fig 1B) was higher in the single-stream task (80.5% ± 0.8% SE) than in the dual-stream task [76.3% ± 0.7% SE; t(29) = 5.51, p<0.001, d = 1.02; paired t-test]. This suggests that comparing two streams was more difficult than averaging a single-stream (of otherwise physically identical inputs; Fig 1A).

To characterize how the numerical value (1–9) of a sample influenced subsequent choice, we calculated model-free decision weights (see *Materials and methods*). In the resulting psychometric function, compression would be characterized by a shallower local slope near extreme values (e.g., 1–2 and 8–9) compared to midrange values (e.g., 4–6). Anti-compression, in contrast, would be characterized by the opposite, a steeper slope near extreme values, compared to midrange values. Descriptively, the weighting curves showed the former (i.e., compression) in the single-stream task (Fig 1C, *left*), and the latter (i.e., anti-compression) in the dual-stream task (Fig 1C, *right*).

For quantitative analysis, we fitted a simple psychometric model (see *Materials and methods*), which characterizes the transformation of sample values (1–9) as a sign-preserving power function with exponent kappa (*k*; where *k*<1 indicates compression and *k*>1 anti-compression). The model, which has been extensively used and validated in previous work [e.g., 4,7,8,10], further includes parameters for overall bias (*b*) towards larger or smaller numbers (*b* > or < 0) and decision noise (*s*, see *Materials and methods*).

The model captured the key characteristics of the psychometric weighting functions in the two tasks well (Fig 1C, cf. black: model and blue/orange: data). Model comparisons using the Akaike information criterion (AIC) confirmed that in both tasks, the data was significantly better described by our nonlinear model (where *k* is a free parameter) than by a linear model with *k* = 1 [single-stream: mean AIC = 254.00±8.05 SE vs. 259.50±6.97 SE, t(29) = -2.06, p = 0.048, d = 0.13; dual-stream: mean AIC = 286.10±6.53 SE vs. 302.55±6.40 SE, t(29) = -6.12, p<0.001, d = 0.46; paired t-tests comparing mean AICs], indicating that the weighting in both tasks was nonlinearly distorted (in opposite ways; see below).

The best-fitting parameter estimates for each task are shown in Fig 1D. Of main interest was parameter kappa (*k*), which indicates the extent to which a weighting policy is compressed (*k* < 1) or anti-compressed (*k* > 1) relative to a linear weighting (*k* = 1). Indeed, *k* was significantly smaller than 1 in the single-stream task [M = 0.82, SE = 0.07, t(29) = -2.43, p = 0.02, d = 0.44; one-sample t-test against 1] and significantly larger than 1 in the dual-stream task [M = 1.82, SE = 0.16, t(29) = 5.21, p<0.001, d = 0.95], which confirms a significant compression in single-stream averaging, and significant anti-compression in dual-stream comparison. The essential difference in transformation (Eq 1) associated with these different *k* values is illustrated in Fig 1C (insets). While the transformation in the single-stream task (*k* = 0.82) is concave (i.e., shallower towards the extremes), the transformation in the dual-stream task (*k* = 1.82) is convex (i.e., steeper towards the extremes). In direct comparison between the two tasks, accordingly, the difference in distortion (*k*) was highly significant [t(29) = -6.19, p<0.001, d = 1.48; paired t-test].

Examining bias (*b*), we found positive values both in the single-stream task [M = 0.02, SE = 0.01, t(29) = 3.52, p = 0.001, d = 0.64] and in the dual-stream task [M = 0.21, SE = 0.06, t(29) = 3.34, p = 0.002, d = 0.61]. Thus, judgments in both tasks were overall biased towards larger numbers, which is consistent with previous work [7,9,10]. Finally, noise (*s*) was

significantly higher in the dual-stream (M = 1.23, SE = 0.07) than in the single-stream task [M = 0.98, SE = 0.06; t(29) = -4.40, p<0.001, d = 0.70; paired t-test]. This is consistent with the lower level of accuracy in the dual-stream task (see Fig 1B).

Together, our experimental manipulation was successful in inducing opposite types of psychometric distortions, while participants processed identical stimuli in the two task conditions [see also 7]. Whereas decision weighting was mildly compressed (i.e., shallower towards extreme values) in single-stream averaging, it was anti-compressed (i.e., steeper towards extreme values) in dual-stream comparison.

## EEG results

**Multivariate (RSA) results.**   Turning to the EEG data, we first examined the encoding of sample information in multivariate ERP patterns using RSA (see *Materials and methods*). Specifically, we examined in each of the two tasks (single-stream and dual-stream) the extent to which RSA patterns encoded (i) the concrete digit that was shown as sample stimulus (e.g., "4" or "8"), (ii) its color (i.e., red or blue), and (iii) the numerical magnitude information in a sample (i.e., 1–9, in terms of a numerical distance effect; see *Materials and methods* and Fig 2A).

The visual attributes of a sample (i.e., its color and digit shape) were encoded early on, from approximately 100 to 700 ms after sample onset, in both tasks (Fig 2B, all $p_{cluster}$<0.001). From approximately 200 ms on, the RSA patterns also encoded the samples' numerical magnitude, in terms of a significant numerical distance effect (single-stream: $p_{cluster}$<0.001; dual-stream: $p_{cluster}$<0.001), thus replicating and extending previous work [9,10,13–15]. Descriptively, the numerical distance effect observed in the single-stream task, while robustly significant, appeared weaker than that in the dual-stream task.

Comparing the RSA time courses between the two task conditions (Fig 2C) confirmed that the numerical distance effect was significantly stronger in the dual-stream task ($p_{cluster}$<0.002). Surprisingly, we found no difference in color encoding between the two tasks, even though color was task-relevant only in the dual-stream task, but not in the single-stream task. Instead, in the single-stream task, we observed a relatively stronger representation of the concrete digit (i.e., the unique number symbol) that had been displayed ($p_{cluster}$<0.005). This effect was evident in a relatively late time window (approximately 400–600 ms, that is, only after the early visual encoding of digits and color).

Together, multivariate ERP patterns in both task conditions robustly encoded information about the sample's color, the number symbol it showed, and its numerical magnitude. The representation of samples in the single-stream task, however, showed qualitative differences, in terms of a relatively weaker encoding of numerical magnitude, and a relatively stronger encoding of the concrete sample stimuli.

**Neurometric RSA results.**   Next, we examined potential distortions of the "neural numberline" underlying the neural magnitude representation disclosed in the above RSA results. To this end, we parameterized the numerical distance model (Fig 2D, *right*) to reflect distortions by *k* (compression/ anti-compression) and *b* (bias towards/against larger numbers), analogously as in our psychometric model (see *Materials and methods*, Eq 1). We then used exhaustive gridsearch to determine for each participant the parameter combination with which the model fitted the data best. Fig 3B illustrates the improvement in fit (in terms of *Δ r* relative to the standard model with *k* = 1 and *b* = 0; cf. Fig 2D, *right*) in a representative time window (0.2–0.6 s; cf Fig 2C and 3A). Note that neurometric mapping was performed using a log scale of *k* (where *log*(*k*) = 0 corresponds to *k* = 1, see Fig 3B) to avoid fitting bias (see *Materials and methods*).

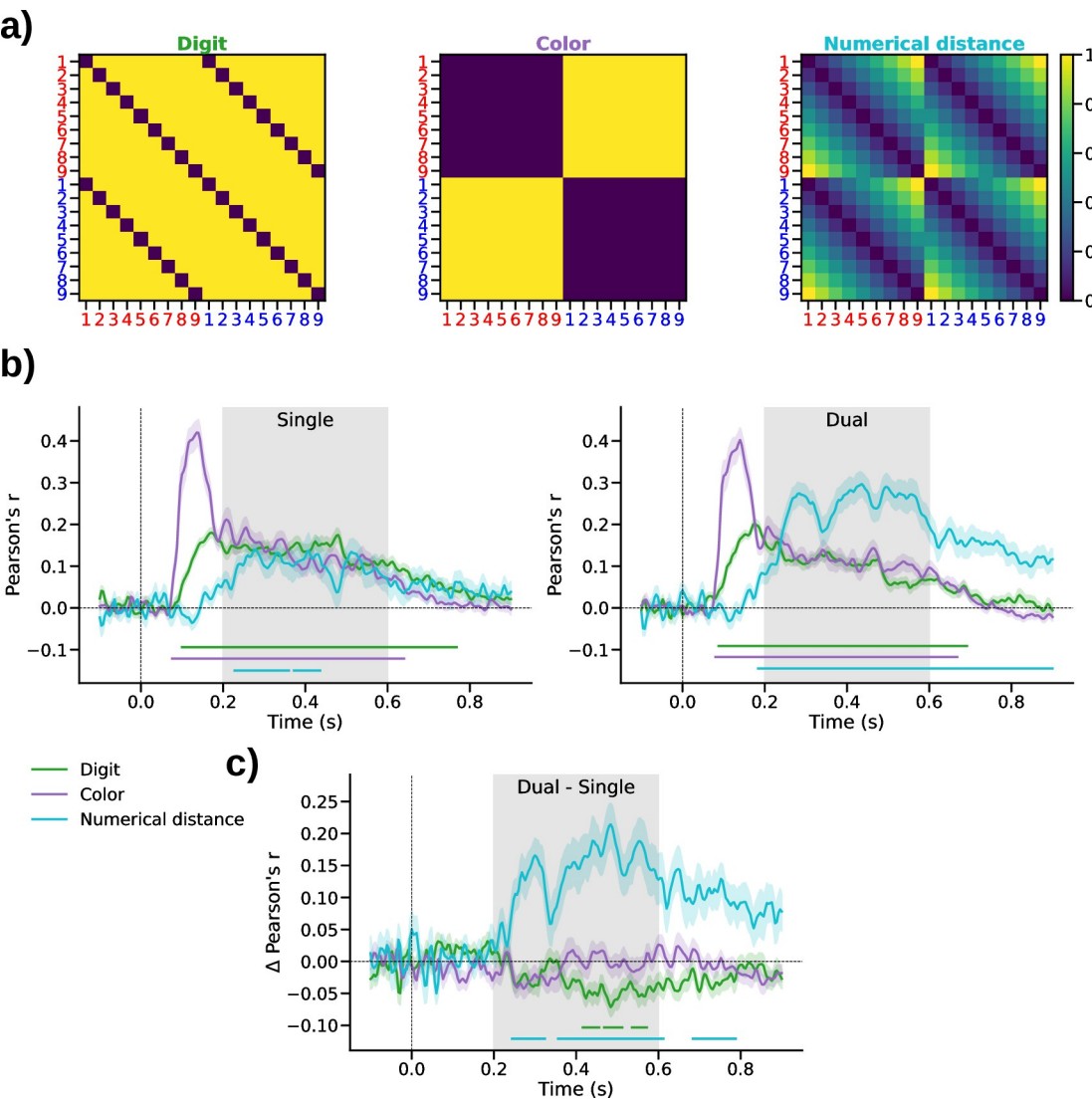

**Fig 2. RSA results. a)** Model RDMs encoding individual sample attributes. *Left*, "Digit" model encoding the unique number symbols; *Middle*, "Color" model encoding whether a sample was red or blue; *Right*, "Numerical distance" model encoding the samples' magnitude (1–9). For visual clarity, the model RDMs are illustrated before orthogonalization (see *Materials and methods*). **b)** Correlation between orthogonalized model RDMs and the empirical ERP-RDMs in the single-stream (left) and dual-stream (right) task. Colored shadings show SE. Marker lines on bottom indicate significant differences from zero ($p_{cluster}<0.005$). Gray shading outlines the time window used in subsequent neurometric analysis (see Fig 3 below). **c)** Difference in correlation between the single- and dual-stream tasks. Same conventions as in *b*.

The results (Fig 3B) replicated our previous finding of a significant distortion of the neural numberline in the dual-stream task (Fig 3B *right*). Specifically, like in our previous work [10,13], we observed neurometric estimates of $k > 1$ (i.e., anti-compression) and of $b > 0$ (i.e., a bias towards larger numbers; p = 0.013, FDR-corrected), which mirrors the pattern observed in the behavioral data (cf. Fig 1C and 1D, *orange*). However, contrary to our expectations, we found no evidence for a neurometric compression in the single-stream task (Fig 3B, *left*), where the psychometric weighting in behavior was clearly compressed (cf. Fig 1C and 1D, *blue*). Descriptively, the neurometric map in the single-stream task indicated a pattern similar to that in the dual-stream task (i.e., anti-compression $k > 1$, and positive bias, $b > 0$). However,

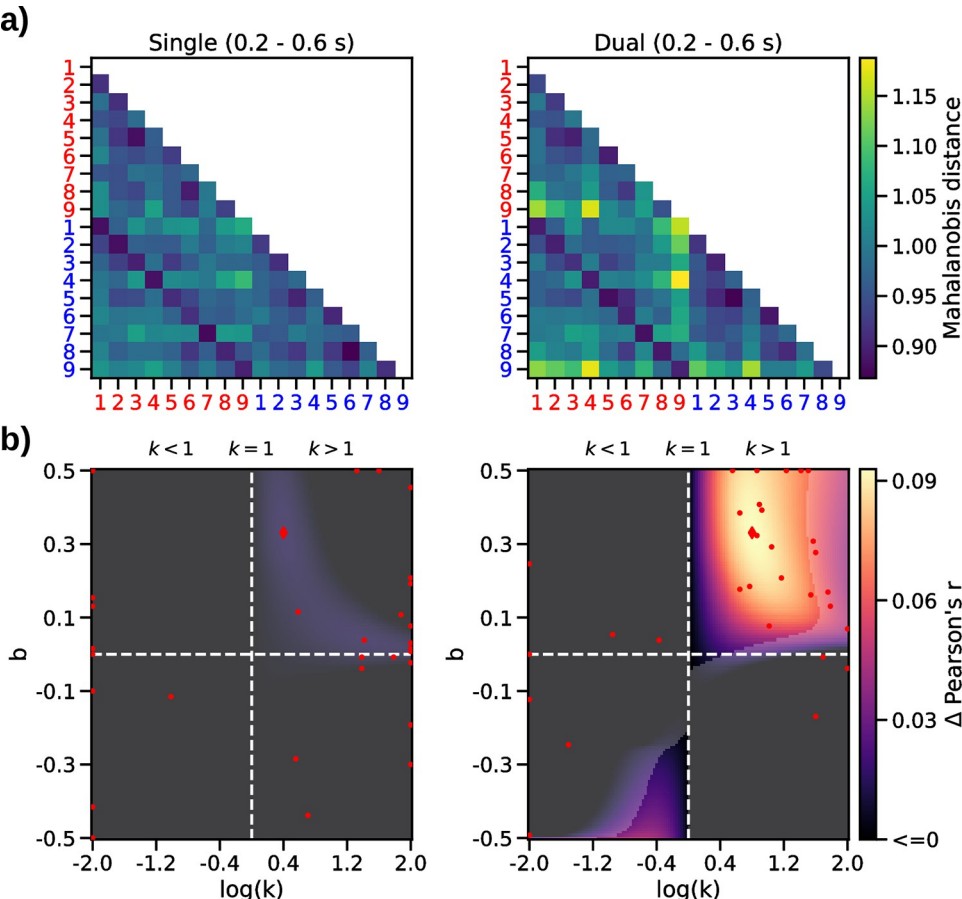

**Fig 3. Neurometric RSA results. a)** Mean ERP-RDMs in the time window of the numerical distance effect (see gray shading in Fig 2B). **b)** Mean neurometric maps. *Left*: single-stream task; *right*: dual-stream task. Dashed lines indicate linear ($k = 1$) and unbiased ($b = 0$) parameterizations. Color scale indicates increase in model correlation ($\Delta r$) relative to the standard model where $k = 1$ and $b = 0$. Transparency mask delineates where the increase was statistically significant (p<0.05, FDR corrected). Red markers show maxima (diamond, mean map; dots, individual participant maps).

the improvement in fit over the linear/unbiased model was weak and statistically not significant (Fig 3B, *left*), potentially reflecting that the numerical distance effect in the single-stream task was overall weaker (cf. Fig 2B).

Examining the mean neurometric parameter estimates in the two tasks statistically, they showed significant anti-compression ($k > 1$) both in the dual-stream [M = 3.05, SE = 0.40; t (29) = 5.16, p<0.001, d = 0.94] and in the single-stream task [M = 3.81, SE = 0.57; t(29) = 4.90, p<0.001, d = 0.89, t-tests against 1]. Direct comparison of neurometric $k$ between the two tasks showed no significant difference [t(29) = 1.25, p = 0.22, d = 0.28, paired t-test]. A positive offset bias ($b$) was evident in the dual-stream task [M = 0.18, SE = 0.04; t(29) = 3.97, p<0.001, d = 0.73] but not in the single-stream task [M = 0.04, SE = 0.05; t(29) = 0.75, p = 0.46, d = 0.14, t-tests against 0; difference between tasks t(29) = -2.27, p = 0.03, d = 0.54, paired t-test]. Together, the neurometric RSA results yielded no evidence for a compression of numerical magnitude akin to that observed in behavior in the single-stream task. Instead, the results indicated anti-compression ($k > 1$) in both tasks, although it should be noted that the improvement in model fit (relative to a linear model) in the single-stream task was small and not statistically significant (Fig 3B, *left*).

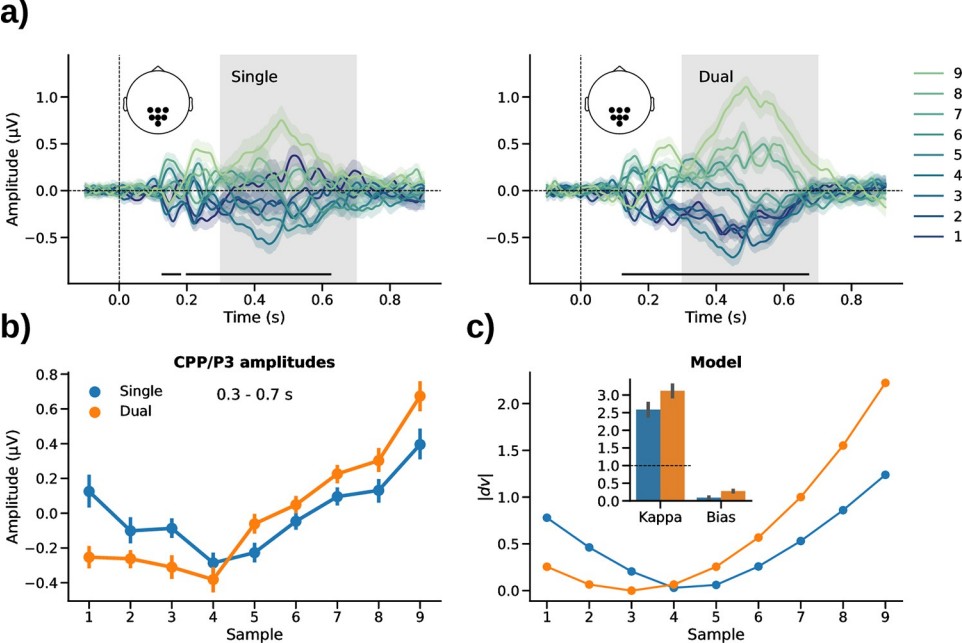

**Fig 4. Univariate CPP/P3 results. a)** Centro-parietal ERPs (mean-subtracted) evoked by sample values 1–9 in the single- (*left*) and dual-stream (*right*) tasks. Marker lines on bottom indicate significant differences between the 9 different sample values ($p_{cluster}<0.001$, repeated measures analysis of variance). **b)** Mean CPP/P3 amplitudes averaged over the time window outlined by gray shading in *a* (0.3–0.7 s). **c)** Neurometric model fit of CPP/P3 amplitudes. Line plot shows grand mean fit. Inset bar graph shows mean parameter estimates. Error indicators in all panels show SE.

**Univariate ERP results (CPP/P3).** We complemented our analysis by examining neurometric distortions also in univariate ERP signals, specifically in the sample-evoked CPP/P3 response. Previous research has implicated the CPP/P3 in decision formation, with its amplitude reflecting the perceived strength of evidence [25–30]. CPP/P3 amplitudes were previously also found to be modulated by numerical sample values in the context of a dual-stream comparison task [10].

We observed modulations of CPP/P3 amplitude by numerical value both in the single-stream (Fig 4A, *left*; $p_{cluster}<0.001$) and in the dual-stream task (Fig 4A, *right*; $p_{cluster}<0.001$, repeated measures analyses of variance), with the modulation in the dual-stream task appearing descriptively stronger. The mean amplitudes showed a U-shaped pattern over numbers 1–9 (Fig 4B), consistent with previous findings that CPP/P3 reflects the strength of decisional evidence in an unsigned fashion, that is, a theoretical quantity similar to the *absolute* $|dv|$ in our psychometric model, which would reflect the strength of evidence for either choice, "<" or ">" [see also 25–30]. Thus, for model-based analysis, we fitted Eq 1 to the CPP/P3 amplitude data (using pairwise distance matrices and gridsearch analogous to our neurometric RSA above), but using $|dv|$ to generate the model-predicted pattern.

The best-fitting parameter estimates are shown in Fig 4C (*inset bar graph*). Mirroring the RSA results, the parameter estimates based on CPP/P3 amplitude showed significant anti-compression ($k > 1$) in both tasks [single-stream: M = 2.58, SE = 0.55; t(29) = 2.88, p = 0.007, d = 0.53; dual-stream: M = 3.12, SE = 0.51; t(29) = 4.15, p<0.001, d = 0.76, t-tests against 1], with no significant difference between tasks [t(29) = -0.69, p = 0.49, d = 0.18, paired t-test]. A positive offset bias was again observed in the dual-stream task [M = 0.28, SE = 0.06; t(29) = 5.00, p<0.001, d = 0.912], but not in the single-stream task [M = 0.09, SE = 0.05; t(29) = 1.78, p = 0.09, d = 0.33, t-tests against 0; difference: t(29) = -2.97, p = 0.006, d = 0.63, paired t-test].

The univariate ERP analysis thus corroborates our RSA finding that the neural signature of number processing in the single-stream task was not characterized by compression but by anti-compression, despite the evidence for compression in subsequent behavioral choice (cf. Fig 1C and 1D).

Together, our findings indicate a lack of correspondence between the psychometric and neurometric distortions observed in our tasks. Possibly consistent with this, explorative analyses also showed no substantial correlations between our participant's individual psycho- and neurometric distortion parameters. This was especially true in the single-stream task (all r <0.26, all p>0.17, uncorrected). Anecdotally, there was an indication of a positive relation between psycho- and neurometric $k$ only in the dual-stream task (RSA: r = 0.32, p = 0.08; ERP: r = 0.41, p = 0.03, uncorrected), which, however, was insignificant after controlling for multiple testing (all p>0.10, FDR-corrected). We exert caution in interpreting these brain-behavior correlations given a relatively small sample size [31].

## Discussion

We observed opposite types of psychometric distortions (compression or anti-compression) in behavior when participants were instructed to process an identical stream of numbers in two distinct ways, namely, comparing the complete stream against a fixed target value or comparing two sub-streams against each other. However, contrary to expectations based on past research, EEG signals associated with the processing of the individual number samples showed evidence for anti-compression (i.e., enhanced processing of extreme values) under both instructions, regardless of whether extreme values were over- or underweighted in subsequent behavioral choice. We further observed qualitative differences between the sample representations in the two tasks, with a relatively weaker encoding of the samples' numerical magnitude in the single-stream task.

In psychophysical research concerned with how attributes of a physical stimulus (e.g., size, weight, color) relate to their subjective experience or perception, human observers are commonly found to underweight extreme values. Such subjective compression of magnitude can be observed in a great variety of settings, from basic sensory-perceptual judgments [2,16] to economic decisions [5,22,23]. There exist various theoretical accounts for the origin and potential benefits of subjective compression in perception and decision making [4,6,32–37]. Our present results seem to contrast these vast literatures with the finding that human EEG signals tended to reflect the magnitude of numerical values in an *anti*-compressed fashion, even when they were mildly compressed in later choice.

While the typical finding in many tasks is compression, there are task contexts where observers *over*weight extreme samples in choice behavior, indicating an anti-compression of sample values [7–12,38,39]. We recently showed that such anti-compression can be beneficial in tasks that are computationally challenging (like our dual-stream comparison task), and where capacity-limited observers may be forced to selectively focus on a subset of the samples at the expense of others [7, see also 40]. Our present findings may suggest that in task contexts in which higher-level processing capacities are exceeded, participant behavior might more directly reflect the brain's default response to extreme values (i.e., privileged processing), which may result in anti-compression.

Despite the lack of evidence for different neural geometries of numerical magnitude in the two tasks, the samples' overall representation in neural signals yet differed. In the dual-stream task, neural signals encoded the numerical magnitude of a sample more strongly than in the single-stream task. This result was unexpected because nominally, the numerical magnitude of the presented samples was equally relevant in both tasks. In the single-stream task, in turn, we

found relatively stronger encoding of which unique number symbol was presented. Further, although the color of a sample (red/blue) was task-irrelevant in the single-stream task, the neural encoding of color was as strong and as sustained as in the color-based dual-stream task. While these results were unexpected, they may suggest more general differences in the role of "abstract" magnitude processing [18,20,41–43] between the two tasks. Potentially, in the more challenging dual-stream task participants relied more directly on an intuitive "sense of magnitude" [44–46] to gauge a sample's decision value. The computationally simpler single-stream task, in contrast, may have allowed them to engage in more symbolic-analytic processing (e.g., approximate arithmetics and/or verbalization)–processes that might be less amenable to EEG-decoding than the numerical distance pattern that prevailed in the dual-stream task.

The evidence for neural anti-compression of sample values in both of the tasks was corroborated by univariate CPP/P3 signals which had previously been implicated in the decisional evaluation of stimulus information [25–30]. Interpreting the amplitude of CPP/P3 signals as an index of the perceived strength of evidence, our findings in the single-stream task show a mismatch between the pattern observed in sample-by-sample processing (anti-compression, as revealed by neural data) and that in eventual judgment of the aggregate stream (compression, as evident in behavior). Future work will be required to identify the neural mechanisms underlying the eventual downweighting of extreme values in such task contexts, despite enhanced encoding in sample-level decision signals. It should be noted that the amplitude of P3 signals is known to be modulated by various factors, including how rare or surprising an event is [47,48]. However, the sample values in our experiment were uniformly distributed, that is, each value occurred equally often on average, ruling out an explanation in terms of stimulus frequency.

How sample values are weighted has major implications for the behavioral choices people make. One and the same numerical evidence, as in the present study, may lead to opposite choices, depending on how much weight people give to extreme values, relative to intermediate values. For instance, anti-compression can explain systematic violations of "rational" axioms, such as transitivity, in multi-attribute choice [40,49]. More generally, nonlinear distortions of objective data (such as numbers) have often been interpreted as paradigmatic manifestations of seemingly "irrational" human behaviors in decision making [e.g., nonlinear probability weighting in risky choice as assumed in cumulative prospect theory; 5]. However, over the past years a new literature has evolved that recasts these behaviors as well-adapted policies of capacity-limited observers, fostering rather than hampering their measurable performance under these constraints [3,40,50–55]. Here, we sought to shed new light on the open question of how such adaptive distortions may arise in neural processing, in terms of the signal patterns evoked by samples of evidence while observers are in the process of reaching a decision. Our finding of an anti-compression of values in EEG patterns—regardless of subsequent behavior—raises the question at which processing stage value compression emerges, and how it leads to, for instance, the well-known "diminishing sensitivity" to larger values in economic choices [e.g., 5].

## Materials and methods

### Ethics statement

All participants provided written informed consent and received €10 per hour as compensation, in addition to a €10 flat fee for participation, as well as a performance-dependent bonus (€7.03 ± 1.08 on average). The study was approved by the ethics committee of the Max Planck Institute for Human Development.

## Participants

Thirty-two healthy volunteers took part in the experiment. We excluded two participants who reported having misunderstood the task instructions and who performed near chance level (50% correct choices) in one of the tasks (52% and 53%, respectively; both p>0.4, Binomial tests against 0.5). Results are reported for the n = 30 remaining participants (15 male, 15 female; mean age 27.4 ± 4.9 years; one left handed).

## Experimental design

Each participant performed two variants of a sequential number integration task [7,10]. The stimulus protocols in the two task variants were identical (Fig 1A). On each trial, participants viewed a sequence of 10 Arabic digits (randomly drawn from a uniform distribution of numbers 1 to 9) displayed in either red or blue font color (randomly assigned to each sample, with the restriction that each sequence contained 5 red and 5 blue samples). In the "averaging" (single-stream) task, participants were asked to judge whether the average of all 10 number samples in the sequence (regardless of their color) was larger or smaller than 5. In the "comparison" (dual-stream) task, participants were asked to compare the red and the blue samples against each other and to indicate which of the two had the higher average value. Thus both tasks required participants to evaluate all 10 number samples. Past behavioral work has shown a psychometric compression of number values in the single-stream task, whereas anti-compression was evident in the latter dual-stream task [7,10].

The experiment was programmed in Python using the PsychoPy package [56] and run on a Windows 10 PC. Throughout the experiment, we additionally recorded eye-movements using an EyeLink 1000 Plus (SR Research Ltd., Canada), which were not analyzed in the present study. Participants were informed about the eye-movement recording and were instructed to keep their gaze at the center of the screen throughout the experiment.

Each trial started with a white central fixation stimulus (a combination of bulls eye and cross hair; Thaler et al. 2013 [57]) on an otherwise black screen. After 500 ms, the fixation stimulus disappeared and the number sequence was presented at a rate of 350 ms per sample (font Liberation Mono; height 3˚ visual angle; see Fig 1A). Each sample was smoothly faded to black after 270 ms to improve the visual experience of the stimulus transitions. After the last sample, participants were prompted to enter a response by pressing the left or right button on a USB response pad (The Black Box ToolKit Ltd., UK). To avoid left/right motor response preparation during sequence presentation, in each of the two tasks, we randomized the mapping of responses ("smaller" or "larger" in the averaging task; "red" or "blue" in the comparison task) onto left/right button presses trial-by-trial, using a response screen (Fig 1A, right).

If participants failed to respond within 3 s, the trial was discarded and after a delay of 100 ms, a message ("too slow!") was displayed in red color for 1 s. On average, participants responded within 0.67±0.27 s and timeouts occurred only on 0.03% of trials. On the remaining trials, performance feedback was displayed ("correct" or "wrong", in green or orange color, respectively) for 350 ms. All feedback was displayed centrally in Liberation Mono font with a height of 1˚ visual angle. On 4.85% of trials, in which the objective sequence average was precisely 5 (in the single-stream task) or identical for red and green samples (in the dual-stream task), a random feedback message was displayed. These trials were excluded from the analysis of accuracy levels, but were included in the modeling- and EEG analyses [see also 10]. After feedback, the central fixation stimulus re-appeared and after 500 to 1500 ms (randomly varied), the next trial started.

Each participant first performed 300 trials in one of the tasks (single-stream averaging or dual-stream comparison), followed by 300 trials in the other task (in counterbalanced serial

order across subjects). Thus, 3000 number samples were presented in each task condition and participant. Trials were performed in blocks of 50, with summary performance feedback (percentage correct choices) being provided after each block. After completing all blocks of the first task, participants received the instructions for the second task. To avoid differences in stimulus input, the second task was performed on the exact same number sequences as the first task. Upon completing the second task, participants received a monetary bonus depending on their mean accuracy in both tasks.

## EEG recording

The experiment was performed in an electrically shielded and soundproof cabin. Scalp EEG was recorded with 64 active electrodes (actiCap, Brain Products GmbH Munich, Germany) positioned according to the international 10% system. Electrode FCz was used as the recording reference. We additionally recorded the horizontal and vertical electrooculogram (EOG) and electrocardiogram (ECG) using passive electrode pairs with bipolar referencing. All electrodes were prepared to have an impedance of less than 10 kΩ. The data were recorded using a BrainAmp DC amplifier (Brain Products GmbH Munich, Germany) at a sampling rate of 1000 Hz, with an RC high-pass filter with a half-amplitude cutoff at 0.016 Hz (roll-off: 6 dB/octave) and low-pass filtered with an anti-aliasing filter of half-amplitude cutoff 450 Hz (roll-off: 24 dB/octave). The dataset is available on GIN in source format and formatted according to the Brain Imaging Data Structure (BIDS) using MNE-BIDS [58–60]: https://doi.org/10.12751/g-node.lir3qw.

## Behavioral data analysis

We calculated model-free decision weights to examine how strongly each numerical sample value (1, 2, . . ., 9) contributed to participants' choices in the two tasks. In the single-stream task, these weights were computed as the proportion of times a sample value was associated with the subsequent choice "larger". Analogously, in the dual-stream task, the weights were computed as the proportion of times the sample's color (i.e., red or blue) was subsequently chosen [see also 10]. For comparison with model predictions (see below), we computed decision weights also from the model-predicted choice probabilities (*CP*, see Eq 3) obtained from using the best fitting parameter estimates in each participant.

## Psychometric model

To quantify psychometric distortions (i.e., compression or anti-compression) in behavior, we used a simple psychometric model that has been used extensively in previous work [4,7,9,10,13]. The model formalizes the transformation of objective sample values *X* (here: numbers 1–9, normalized to the range [–1, 1]) into a subjective decision value *dv* as a sign-preserving power function:

$$dv = \frac{X + b}{|X + b|} \times |X + b|^{k}, \tag{1}$$

where exponent *k* (kappa) determines the overall shape of the transformation ($k<1$: compression; $k = 1$: linear; $k>1$: anti-compression). Parameter *b* (bias) implements an overall weighting bias towards smaller ($b<0$) or larger ($b>0$) numbers. Sample-level decision values (*dv*) are integrated into a trial-level decision value (*DV*) by summation over samples:

$$DV = \sum_{i=1}^{10} dv_{i} \times c_{i} \tag{2}$$

where $c$ is an indicator variable denoting a sample's color (red: -1, blue: +1) in the dual-stream task. In the single-stream task, $c$ was fixed at 1. This way, Eq 2 effectively implements a comparison between streams in the dual-stream task, and simple averaging in the single-stream task. Finally, the trial level decision value ($DV$) is transformed into a choice probability according to a logistic function:

$$CP = \frac{1}{1 + e^{\frac{-DV}{s}}},$$

(3)

where $CP$ denotes the probability of choosing ">5" (in the single-steam task) or "blue>red" (in the dual-stream task), and parameter $s$ quantifies the level of decision noise, with larger values of $s$ implying more random choices.

The model was fitted to each participant's individual choice data using the Nelder-Mead method as implemented in SciPy [61], with parameter values restricted to the ranges ($k$: [0, 5]; $b$: [-0.5, 0.5], $s$: [0.01, 3]). Fitting was performed iteratively using 900 combinations of different starting values for each task condition, and the solution with the lowest AIC was used in the analysis. Statistical analysis of the fitted parameters proceeded with conventional inferential tests on the group level.

In previous studies using this model, we also included a "leakage" parameter [30] to additionally account for recency effects in decision weighting [7,10]. We here included no such additional parameter in the interest of parsimony, as our focus was on the overall correspondence of distortions $k$ between EEG and behavior. Control analysis including a leakage parameter in the behavioral model fully replicated the present results for parameter $k$ [see also 7]. We report for completeness that the leakage parameter was larger in the dual- than in the single-stream task (p<0.01), possibly related to the dual-stream task being overall harder [for detailed discussion of the processing demands in single- and dual stream tasks, see 7].

## EEG preprocessing

We used functions from MNE-Python [62] and PyPrep [63, based on 64] to automatically mark noisy segments and bad channels in the EEG recordings. We additionally screened all recordings visually to reject noisy segments or bad channels that the automatic procedures had missed. This way, on average, 2.5 ± 1.6 channels were discarded per participant. Next, we corrected ocular and cardiac artifacts using independent component analysis (ICA). To this end, we high-pass filtered a copy of the raw data at 1 Hz and downsampled it to 100 Hz. We then ran an extended infomax ICA on all EEG channels and time points that were not marked as bad in the prior inspection. Using the EOG and ECG recordings, we identified stereotypical eye blink, eye movement, and heartbeat artifact components through correlation with the independent component time courses. We visually inspected and rejected the artifact components before applying the ICA solution to the [65]. We then filtered the ICA-cleaned data between 0.1 and 40 Hz, interpolated bad channels, and re-referenced each channel to the average of all channels.

## Event-related potentials (ERPs)

We epoched the preprocessed data from −0.1 to 0.9 s relative to each sample stimulus onset. Remaining bad epochs were rejected using a thresholding approach from the FASTER pipeline [Step 2; 66]. On average, n = 5764 clean epochs (96.1%) per participant were retained for analysis. The epochs were then downsampled to 250 Hz and baseline corrected relative to the period from −0.1 to 0 s before stimulus onset. Since our analyses focused on stimulus-specific effects, we subtracted the overall mean waveform from the individual epochs, in each of the

two task conditions. The mean-subtracted epochs were then averaged into stimulus-specific ERPs for each sample value (1, 2,. . ., 9) in each color (red/blue). Note that the individual samples in a stream were statistically independent by design, allowing us to examine stimulus-specific ERP responses in a time window that overlapped with the onset of the next sample stimulus.

## Representational similarity analysis (RSA)

We used representational similarity analysis [RSA; 67] to examine the encoding of sample information in multivariate ERP patterns. Specifically, we examined the representational geometry of our stimulus space (numbers 1 to 9, colored red or blue) in terms of the multivariate (dis-)similarity between the ERP topographies (64 channels) associated with the 18 different stimuli. Representational dissimilarity was computed as the Mahalanobis distance, between each pair of stimuli, yielding an 18x18 representational dissimilarity matrix (RDM), at each time point of the peri-sample epoch. To compute the Mahalanobis distance, we fitted a general linear model to the z-scored trial data, with each stimulus type specified as a condition, and used the residual trial-by-trial variance for pairwise distance calculation. This procedure ensured multivariate noise normalization for the RDMs [68]. Below, we refer to the thus obtained RDMs as ERP-RDMs.

To examine the information encoded in the ERP-RDM time courses, we used three different model RDMs (see Fig 2A) reflecting (i) the unique digit symbols, with minimum dissimilarity between identical digits, and maximum dissimilarity between distinct digits ("digit" model), (ii) the samples' color, with minimum (maximum) dissimilarity between same (different) colors ("color" model), and (iii) the numerical distance between samples, that is, the arithmetic difference between their objective number values ("numerical distance" model). To render the three models fully independent, we recursively orthogonalized each model RDM with respect to all others using the Gram-Schmidt process [10,13]. Finally, we assessed the match between each model RDM and the empirically observed ERP-RDMs via Pearson correlation at each time point, using only the lower triangle of the RDMs and omitting the diagonal, to exclude redundant matrix cells.

For statistical analysis of the RSA time courses, we used t-tests against zero with cluster-based permutation testing to control for multiple comparisons over time points [69]. To test whether RSA results differed between the single- and dual-stream tasks, we first computed their difference, followed by cluster-based permutation tests against zero. All permutation tests were performed over 1000 iterations with a cluster-defining threshold of $p = 0.01$ and cluster length as the critical statistic (thresholded at $p = 0.01$).

## Analysis of neurometric distortions

The theoretical model underlying conventional RSA of numerical distance effects (see above) is a straight number line, where the numbers (1, 2,. . ., 9) are equidistant. The standard numerical distance model (Fig 2A, *right*) is equivalent to a model of *dv* according to Eq 1 (see *Psychometric model*) where $k = 1$ and $b = 0$. To examine potential nonlinear distortions of the number representations in neural signals ("neurometric" distortions), we constructed numerical distance models based on *dv* while varying the values of $k$ (from $log(k) = -2$ to $+2$, see Fig 3B) and $b$ (from -0.5 to 0.5). Varying $k$ on a log scale centered around $log(k) = 0$ (i.e., $k = 1$) ensured that parameter estimates were not biased to show anti-compression (or compression) by chance in subsequent gridsearch. For each parameter combination, we correlated the resulting model RDMs with the ERP-RDM, yielding a grid ("neurometric map") of the parameter space (see Fig 3B). The parameter combination with the maximum correlation was used as the

estimate of the participant's neurometric distortion parameters (see also Spitzer et al., 2017; Appelhoff et al., 2022). Statistical analysis of the neurometric parameter estimates proceeded with conventional statistical tests on the group level.

## Univariate ERP analysis

For complementary inspection of univariate ERP responses evoked by the number samples, we examined the stimulus-specific ERP (see above) for each sample value (1–9; collapsed across red/blue colors). To focus on CPP/P3 responses (see *Results*), the ERPs were pooled over centro-parietal channels (CP1, P1, POz, Pz, CPz, CP2, P2) and amplitudes were examined in a time window from 300 ms to 700 ms based on previous work [10,13,30,70]. The ERP time courses were analyzed statistically using cluster-based permutation testing (see above). For model-based analysis, we used the same approach as in our analyses of neurometric distortions in RSA (see above), except that the model RDMs were constructed from $|dv|$ (i.e., the absolute, unsigned magnitude of $dv$, see Results) and correlated with the pairwise differences in univariate ERP amplitude between samples 1–9.

## Acknowledgments

We thank Anna Faschinger, Gabriele Inciuraite, Aleksandra Zinoveva, Larissa Samaan, Simon Ciranka, and Jann Wäscher for help with data collection, and Verena Clarmann von Clarenau and Thorsten Pachur for helpful discussions and input.

## Author Contributions

**Conceptualization:** Stefan Appelhoff, Bernhard Spitzer.

**Data curation:** Stefan Appelhoff.

**Formal analysis:** Stefan Appelhoff, Bernhard Spitzer.

**Investigation:** Stefan Appelhoff.

**Methodology:** Stefan Appelhoff, Bernhard Spitzer.

**Project administration:** Stefan Appelhoff, Bernhard Spitzer.

**Resources:** Ralph Hertwig, Bernhard Spitzer.

**Software:** Stefan Appelhoff.

**Supervision:** Bernhard Spitzer.

**Validation:** Stefan Appelhoff.

**Visualization:** Stefan Appelhoff, Bernhard Spitzer.

**Writing – original draft:** Stefan Appelhoff, Bernhard Spitzer.

**Writing – review & editing:** Stefan Appelhoff, Ralph Hertwig, Bernhard Spitzer.

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
