## [Decision Letter · Decision Letter 0]

20 Jul 2022

Dear Appelhoff,

Thank you very much for submitting your manuscript "EEG-representational geometries and psychometric distortions in approximate numerical judgment" for consideration at PLOS Computational Biology.

As with all papers reviewed by the journal, your manuscript was reviewed by members of the editorial board and by several independent reviewers. In light of the reviews (below this email), we would like to invite the resubmission of a significantly-revised version that takes into account the reviewers' comments.

Two reviewers have read your paper, and although they see some merit in it, they also have major reservations. I encourage you to have a serious look at these concerns and try to address them.

We cannot make any decision about publication until we have seen the revised manuscript and your response to the reviewers' comments. Your revised manuscript is also likely to be sent to reviewers for further evaluation.

Sincerely,

Marieke Karlijn van Vugt, PhD

Associate Editor

PLOS Computational Biology

Samuel Gershman

Deputy Editor

PLOS Computational Biology

Reviewer's Responses to Questions

**Comments to the Authors:**

Reviewer #1: Here the authors examine how sequentially presented numerical information is encoded in a judgment (i.e, the average of the stream was larger or smaller to a reference value) and in a choice context (i.e., the blue stream was larger on average than the red stream). Although behavioural analyses indicate that in the judgment context people compress numerical information (akin to a concave representation) and that in the choice context the follow the opposite pattern (akin to a convex representation), EEG encoding analyses reveal an “anti-compression” (convex representation) in both contexts.

The paper deals with an interesting research question. Furthermore, the analyses and description of the results are rigorous and clear. I have the following concerns/ suggestions:

1) The authors claim that in the judgment context representation is compressed. But Figure 1c speaks for a rather linear representation, at least at the average level. The model fits (Figure 1d) deviate significantly from linearity. It is not clear if the psychometric model misfits by large in some cases, giving rise to a significant k<1. Showing the decision weights per participant and performing some parameter recovery exercises will help. Additionally, can the authors identify critical trials in which the “compressed” and “anti-compressed” models make opposite predictions? This will help to further corroborate the claim that k<1 in judgment and k>1 in choice.

2) Does the order of presentation influence behaviour, giving rise to primacy/ recency patterns? Additionally the authors state that “as expected, means choice accuracy was higher in the single-stream task…This suggests that comparing two streams was more difficult than averaging a single stream (of otherwise physically identical inputs)”. It is not clear if the authors suggest that judgment is easier than choice. Decisions in the former are made on the basis of 10 samples whilst in the latter on the basis of 5 samples. Assuming some internal noise, the accuracy difference is not surprising. Please clarify that point and perhaps see if having the same level of internal noise (on a per sample basis) in both tasks, explains away the accuracy difference without necessitating a different value of softmax noise.

3) The psychometric and “neurometric” analyses do not agree. This mismatch can be an important finding, or more simply it could be the case that either of these analytical approaches is inaccurate. Importantly, the authors do not establish that the kappa & bias parameters obtained from the behavioural data correlate with the parameters obtained from the RSA analyses. Is there any association between neurometric and psychometric parameters? This comment also relates to point 1) (can the authors back up the kappa><1 conclusion using complementary behavioural analyses?).

Reviewer #2: In this study the authors analyze human behavioral and neural responses to numbers presented in streams when subjects evaluate their average and are asked to either compare it to a reference (task 1) or to the average of another stream (task 2).

Results report a very weak evidence for two different psycho-metric distortions (compression or anti-compression) across the two tasks in the behavioral data, but a very strong evidence for the same neuro-metric distortion (anti-compression) across tasks in the neural data. The two sets of results are presented as opposite/incongruent and so they are discussed.

I think that the results are presented and discussed in a slightly distorted fashion, one that is not fully supported by the data:

1. Behavioral data: the authors claim that the results reported confirm “robust compression in single-stream averaging and robust anti-compression in dual stream comparison”. First, the evidence for compression in the single-stream is far from robust: indeed it is based on a quite low statistic (p= 0.02), at least compared to the evidence for anti-compression in the dual task (p<0.001). Second, I could find a report of the goodness of fits of the model used to evaluate compression/anti-compressions, and from eye inspection of the plots, the single stream model might have a rather poor fit of the data. Third, what one should really compare to decide for compression vs. anti-compression are especially the extreme values. Again, from visual inspection of the data plotted, while for the dual stream it seems clear that the extreme values are overweighed, in the single stream there is extremely little evidence that the extremes are overweighed. Thus, I am far from convinced that these bahavioural data are so clear-cut different across stream-types as the authors would want to report. The authors should provide more evidence for their claim for such a difference.

2. Neural data: the results are repeatedly reported in negative statements (e.g., “the neural processing of number samples in single-stream averaging was not characterized by compression), while the positive results are preceded by minimizing expressions (e.g. “but—if anything—by anti-compression”). However, the picture is not as nuanced as the authors would like to present it: for the case of the ERP analyses, for example, the results clearly and very significantly indicate that the neural processing of numbers for single AND dual streams are both and equally characterized by anti-compression.

3. Discussion: the discussion is centered on the difference between the behavioural and the neural data. For the reasons reported above (point 1), I am not convinced that after all they are so different. I think that the results session and the discussion of this paper should be deeply revised.

**Have the authors made all data and (if applicable) computational code underlying the findings in their manuscript fully available?**

Reviewer #1: Yes

Reviewer #2: **No: **

PLOS authors have the option to publish the peer review history of their article (what does this mean?). If published, this will include your full peer review and any attached files.

Reviewer #1: No

Reviewer #2: No
---

## [Decision Letter · Decision Letter 1]

18 Oct 2022

Dear Appelhoff,

Thank you very much for submitting your manuscript "EEG-representational geometries and psychometric distortions in approximate numerical judgment" for consideration at PLOS Computational Biology.

As with all papers reviewed by the journal, your manuscript was reviewed by members of the editorial board and by several independent reviewers. In light of the reviews (below this email), we would like to invite the resubmission of a significantly-revised version that takes into account the reviewers' comments.

While this revision has much improved, one of the reviewers still has significant concerns about the paper, and suggests a few analyses that can be gone to clarify ambiguities. I therefore recommend a major revision to address these comments.

We cannot make any decision about publication until we have seen the revised manuscript and your response to the reviewers' comments. Your revised manuscript is also likely to be sent to reviewers for further evaluation.

Sincerely,

Marieke Karlijn van Vugt, PhD

Section Editor

PLOS Computational Biology

Samuel Gershman

Section Editor

PLOS Computational Biology

While this revision has much improved, one of the reviewers still has significant concerns about the paper, and suggests a few analyses that can be gone to clarify ambiguities. I therefore recommend a major revision to address these comments.

Reviewer's Responses to Questions

**Comments to the Authors:**

Reviewer #1: The authors have done a good job in addressing most of my comments.

Point 1): I am fully covered by the reply of the authors. One point that the authors have not touched upon is whether there can be some diagnostic trials (for instance low vs. high variance in the comparison task) that could provide additional corroboration for the shape of the distortion.

Point 2): thanks for clarifying the task details. What has not been addressed is the question of serial position effects. What I am missing is a more accurate characterisation of behaviour in the two tasks. Notably, participants find averaging easier than comparison. What could explain this result asides from assuming that the former has a lower noise level than the latter (which is a re-description of the accuracy difference)? My suggestion is to estimate the "leak" of accumulation and compare across tasks (e.g. Wyart, V., Myers, N. E., & Summerfield, C. (2015). Neural mechanisms of human perceptual choice under focused and divided attention. Journal of neuroscience, 35(8), 3485-349).

Assuming that fitting a "noise, leaky" accumulator reveals processing differences in the two tasks, then the question is how these differences map onto the parameter of the psychometric model and especially the distortion parameter. That is, if simulated data are generated by two leaky accumulators with linear input repreresentation but different noise/ leak parameters, is the psychometric fit going to (mistakenly) reveal differences in the distortion parameters? This is largely a sense-checking step that would increase confidence in the reported behavioural results.

3) I appreciate the additional analyses performed. I recommend reporting the absence of correlations (even briefly) since this strengthens the argument that the author make on the missing link between neural representations and behaviour. Given that the behavioural and neural analyses are central in this paper, exploring the relationship between the two seems an obvious step (despite the potential lack of statistical power).

**Have the authors made all data and (if applicable) computational code underlying the findings in their manuscript fully available?**

Reviewer #1: Yes

PLOS authors have the option to publish the peer review history of their article (what does this mean?). If published, this will include your full peer review and any attached files.

Reviewer #1: No
---

## [Decision Letter · Decision Letter 2]

18 Nov 2022

Dear Appelhoff,

We are pleased to inform you that your manuscript 'EEG-representational geometries and psychometric distortions in approximate numerical judgment' has been provisionally accepted for publication in PLOS Computational Biology.

Best regards,

Marieke Karlijn van Vugt, PhD

Section Editor

PLOS Computational Biology

Samuel Gershman

Section Editor

PLOS Computational Biology

Thank you very much submitting your revised version. Both the reviewer and myself think you have fully addressed all concerns and I am happy to now accept the paper. Congratulations!

Reviewer's Responses to Questions

**Comments to the Authors:**

Reviewer #1: The reviewers have done an excellent job addressing my remaining comments. I recommend acceptance of this paper.

**Have the authors made all data and (if applicable) computational code underlying the findings in their manuscript fully available?**

Reviewer #1: Yes

PLOS authors have the option to publish the peer review history of their article (what does this mean?). If published, this will include your full peer review and any attached files.

Reviewer #1: No

---

## [Editor Report · Acceptance letter]

30 Nov 2022

PCOMPBIOL-D-22-00680R2 

EEG-representational geometries and psychometric distortions in approximate numerical judgment

Dear Dr Appelhoff,

I am pleased to inform you that your manuscript has been formally accepted for publication in PLOS Computational Biology. Your manuscript is now with our production department and you will be notified of the publication date in due course.

With kind regards,

Zsofia Freund
